# Role of BAL and Serum Krebs von den Lungen-6 (KL-6) in Patients with Pulmonary Fibrosis

**DOI:** 10.3390/biomedicines12020269

**Published:** 2024-01-24

**Authors:** Piera Soccio, Giorgia Moriondo, Miriana d’Alessandro, Giulia Scioscia, Laura Bergantini, Sara Gangi, Pasquale Tondo, Maria Pia Foschino Barbaro, Paolo Cameli, Elena Bargagli, Donato Lacedonia

**Affiliations:** 1Department of Medical and Surgical Sciences, University of Foggia, 71122 Foggia, Italy; giorgia.moriondo@unifg.it (G.M.); giulia.scioscia@unifg.it (G.S.); laura.bergantini@unisi.it (L.B.); pasquale.tondo@unifg.it (P.T.); mariapia.foschino@unifg.it (M.P.F.B.); donato.lacedonia@unifg.it (D.L.); 2Respiratory Diseases and Lung Transplantation Unit, Department of Medical and Surgical Sciences & Neuro-Sciences, University of Siena, 53100 Siena, Italy; mirian.dalessandro@unisi.it (M.d.); sara.gangi@student.unisi.it (S.G.); paolo.cameli@unisi.it (P.C.); bargagli2@gmail.com (E.B.); 3Institute of Respiratory Diseases, Policlinico Riuniti of Foggia, 71122 Foggia, Italy

**Keywords:** IPF, KL-6, progression, progressive fibrosis, non-progressive fibrosis, biomarkers

## Abstract

**Background:** Interstitial lung diseases (ILDs) encompass a diverse group of disorders affecting the lung interstitium, leading to inflammation, fibrosis, and impaired respiratory function. Currently, the identification of new diagnostic and prognostic biomarkers for ILDs turns out to be necessary. Several studies show the role of KL-6 in various types of interstitial lung disease and suggest that serum KL-6 levels can be used as a prognostic marker of disease. The aim of this study was to analyze KL-6 expression either in serum or bronchoalveolar lavage samples in order to: (i) make a serum vs. BAL comparison; (ii) better understand the local behavior of fibrosis vs. the systemic one; and (iii) evaluate any differences in patients with progressive fibrosis (PPF) versus patients with non-progressive fibrosis (nPPF). **Methods:** We used qRT-PCR to detect KL-6 expression both in serum and BAL samples. Mann–Whitney’s U test was used to compare the differential expression between groups. **Results:** In serum, KL-6 is more highly expressed in PPF than in non-progressive fibrosis (*p* = 0.0295). This difference is even more significant in BAL (*p* < 0.001). Therefore, it is clear that KL-6 values are related to disease progression. Significant differences were found by making a comparison between BAL and serum. KL-6 was markedly higher in serum than BAL (*p* = 0.0146). **Conclusions:** This study identifies KL-6 as a promising biomarker for the severity of the fibrosing process and disease progression in ILDs, with significantly higher levels observed in PPF compared to nPPF. Moreover, the marked difference in KL-6 levels between serum and BAL emphasizes its potential diagnostic and prognostic relevance, providing enlightening insights into both the local and systemic aspects of ILDs.

## 1. Introduction

Interstitial lung diseases (ILDs) constitute a diverse group of lung disorders characterized by interstitial inflammation and/or fibrotic pulmonary processes. They encompass a wide range of underlying causes and clinical presentations, ranging from mild, stable conditions to rapidly progressive and fatal diseases [1].

The progression of ILDs in clinical terms displays significant variability and unpredictability, as some individuals undergo a stable or gradual decline while others experience rapid progression or frequent exacerbations. This diversity poses a challenge to accurately foreseeing the severity and course of the disease [2,3].

ILDs pose a significant challenge due to their unpredictable prognosis and limited diagnostic tools. While established methods such as pulmonary function tests (PFTs), chest X-rays, and High-Resolution Chest Computed Tomography (HRCT) are useful for initial diagnosis, they lack sensitivity and specificity as serial prognostic indicators [3]. Additionally, frequent scans expose patients to increased radiation, and lung biopsies, while informative, are invasive and impractical. Moreover, environmental and cultural factors hinder ILD diagnosis in low- to middle-income countries [4].

Recent research has identified several biomarkers, including monocytes, CA19-9, CA-125, and others, that hold promise in predicting prognosis in ILD patients [5,6,7].

Despite significant strides in medical research, the pathogenesis of IPF remains shrouded in mystery, with a dearth of biomarkers capable of reliably predicting the intensity of disease activity and its subsequent trajectory [8]. Unraveling the underlying pathogenetic mechanisms assumes paramount importance, holding the key to comprehending the hitherto unknown triggers of the condition and paving the way for the development of targeted therapeutic strategies. Recent scientific inquiries have, therefore, embarked on endeavors to dissect the principal mechanisms culminating in the genesis of interstitial fibrosis in IPF. Simultaneously, these investigations seek to identify biomarkers that have efficacy in gauging the degree of disease activity and progression [9,10,11].

On the continuum of these explorations, though still in their embryonic stages, studies have cast a spotlight on KL-6 as a promising biomarker in the assessment of IPF progression [12,13].

Krebs von den Lungen-6 (KL-6) is a high-molecular-weight (200 kDa) sialylated glycoprotein (sialoglycoprotein), present in human mucin-type 1 (MUC1) and expressed on the extracellular membrane of alveolar epithelial cells and bronchial epithelial cells in the regenerative phase [14].

Upon the infliction of damage to the epithelial cell, KL-6 embarks on a journey into the circulatory stream, where its presence becomes quantifiable. In this realm, KL-6 transforms into a chemotactic factor, orchestrating the migration, proliferation, and survival of lung fibroblasts [15]. Recent years have witnessed KL-6 ascend as a sensitive biomarker, especially attuned to instances of alveolar epithelial cell type II (AEC-II) damage or proliferation.

Indeed, the serum levels of KL-6 exhibit a notable elevation in individuals grappling with an array of interstitial lung diseases, a cohort that prominently includes those bearing the burden of idiopathic pulmonary fibrosis [16,17]. The nuances in its serum concentration emerge as harbingers of a graver prognosis and a hastened deterioration of the disease [18], suggesting the potential role of KL-6 as a prognostic and disease progression biomarker.

Furthermore, KL-6 assumes the role of a predictive factor for the onset of exacerbations [19], enhancing its utility as an invaluable indicator of response to treatment. This characteristic lends credence to its role in the monitoring of anti-fibrotic therapy [20,21].

Therefore, with these assumptions in mind, in our study, we first analyzed the expression profile of KL-6 in bronchoalveolar lavage (BAL) and in the serum of patients with progressive pulmonary fibrosis (PPF) and non-progressive pulmonary fibrosis (nPPF) to specify both local and systemic behavior of fibrosis and to assess any differences in expression of this glycoprotein between progressive versus non-progressive forms of fibrosis.

This investigative endeavor seeks to delineate not only the local comportment but also the systemic behavior of fibrosis, aiming to discern potential distinctions in the expression patterns of KL-6 between the cohorts tethered to progressive versus non-progressive forms of fibrosis. Thus, the overarching objective of our study was to furnish a nuanced evaluation of the progression experienced by individuals grappling with pulmonary fibrosis, employing KL-6 levels as a barometer in both serum and BAL analyses.

In undertaking this effort, we aimed to illuminate any disparities emerging at local and systemic levels, thereby reinforcing KL-6’s credentials as a promising and innovative biomarker delineating the course of the ailment. This academic pursuit, supported by the aforementioned assumptions, not only seeks to deepen our understanding of pulmonary fibrosis dynamics but also to confer upon KL-6 the status of a versatile and potent biomarker with implications extending across both BAL and serum analyses. As we navigate the intricate labyrinth of pulmonary fibrosis, our mission prioritizes the pursuit of precision and enlightenment, with KL-6 positioned as a guiding light illuminating the path toward enhanced comprehension and more effective clinical management.

Therefore, this study aimed to assess the progression of pulmonary fibrosis patients by examining KL-6 levels in both serum and bronchoalveolar lavage (BAL). The goal was to emphasize any disparities identified at both local and systemic levels and to recognize KL-6 as a promising new biomarker of progression, applicable in both BAL and serum analyses.

## 2. Materials and Methods

### 2.1. Population

Based on disease progression, 39 patients diagnosed with progressive pulmonary fibrosis (PPF) and 58 patients suffering from non-progressive pulmonary fibrosis who came for observation at the S.C. of Respiratory System Diseases of the Policlinico “Riuniti” of Foggia and the Reference Center for Rare Pulmonary Diseases of the University of Siena between 2015 and 2022 were included in this study. All patients were informed in advance about the purpose and methods of the examinations to which they would be subjected, and written informed consent was obtained from all study participants. Ethical approval was obtained from two institutional review boards (the Ethics Committee - Policlinico Riuniti di Foggia and the Regional Ethics Committee of Siena). All protocols used were in accordance with the principles of the Declaration of Helsinki. For ethical reasons, no healthy donors were enrolled in this study. The diagnosis of various diffuse interstitial lung diseases (ILDs) was determined in accordance with the guidelines provided by ATS/ERS/JRS/ALT in 2022 [22]. The subdivision based on disease progression was formulated in accordance with the criteria recently proposed by Hambly et al. to define disease progression [23]. Specifically, the following were afferent to the progressive pulmonary fibrosis group: 23 individuals affected by idiopathic pulmonary fibrosis (IPF), 5 with hypersensitivity pneumonitis (HP), 3 with non-specific interstitial pneumonia (NSIP), 1 with pulmonary fibrosis in the context of rheumatologic diseases, and 5 individuals suffering from pulmonary interstitiopathy not otherwise specified. Similarly, patients with non-progressive pulmonary fibrosis were classified as follows: 16 patients with idiopathic pulmonary fibrosis that does not progress or progress slowly; 27 individuals with sarcoidosis; 4 with hypersensitivity pneumonitis (HP); 5 with non-specific interstitial pneumonia (NSIP); and 8 with other interstitial lung diseases.

All enrolled subjects underwent respiratory function tests (PFTs), a 6 min walk test (6MWT), and bronchoscopy with bronchoalveolar lavage (BAL) after written consent for the procedure was obtained. Fasting sera and BALs from all patients were collected, aliquoted, and stored at −80 °C until use.

### 2.2. Pulmonary Function Tests (PFT)

The evaluation of pulmonary function involved the utilization of the VMAX22 ENCORE spirometer and V62J plethysmographic booth supplied by VYAIRE-USA (Yorba Linda, CA, USA), which was equipped with VMAX Carefusion software. The reporting adhered to the ATS/ERS 1993 reference guidelines for theoretical values [24]. Employing a plethysmographic technique, comprehensive spirometry encompassed the measurement of both static and dynamic lung volumes, facilitating the identification of ventilatory deficits. The examined parameters included forced expiratory volume in the first second (FEV1), forced vital capacity percent (FVC%), a commonly employed metric for monitoring progression [22], and total lung capacity (TLC). In addition to the aforementioned examination, an alveolus-capillary carbon monoxide diffusion test (DLCO) was performed to assess the transfer of respiratory gases.

### 2.3. Six-Minute Walking Test (6MWT)

All patients underwent a walk test. The test was conducted in a straight corridor featuring a firm walking surface measuring 30 m in length. Each participant was instructed to walk at their own pace for a duration of 6 min, thereby autonomously determining the intensity of the effort and having the flexibility to slow down or halt during the walk. Throughout the test, various parameters were monitored, including heart rate, blood pressure, peripheral oxyhemoglobin saturation, dyspnea assessed by the Borg scale, and the distance covered in meters. The walking test yields submaximal exertion results, reflecting the patient’s exercise capacity [22], with the distance covered serving as a valuable parameter for evaluating disease progression [23].

### 2.4. Bronchoscopy with Broncho-Alveolar Lavage

All patients underwent bronchoscopy with broncho-alveolar lavage for diagnostic purposes under moderate sedation or general anesthesia, according to the guidelines of the BAL Task Force Group of the European Respiratory Society [25]. In short, broncho-alveolar lavage is an ancillary technique of fibrobronchoscopy that allows the collection of cellular and acellular elements from the epithelial surface of the most distal segments of the respiratory tree. The procedure involves the instillation, through the bronchoscope, of 3–5 aliquots of 20–50 mL of saline into the peripheral lung and the subsequent retrieval of at least 30 percent of the volume by gentle aspiration to avoid traumatization or collapse of the distal airways [26]. Part of the lavage fluid thus collected was allocated to this study of lung cellularity; the remaining portion was divided into aliquots and preserved at −80 °C for future analysis.

### 2.5. RNA Extraction

RNA extraction from both serum and BAL samples was performed using the standard TRIzol reagent method (Thermo Fisher Scientific, Waltham, MA, USA), following the manufacturer’s recommended protocol. Subsequently, RNA concentration and quality were evaluated using the NanoDrop 1000 spectrophotometer (Thermo Fisher Scientific). The absorbance ratio at OD_260_/OD_280_ was employed to assess RNA purity.

### 2.6. Quantitive Real-Time PCR (qRT-PCR)

Total RNA was transcribed into cDNA utilizing the iSCRIPT cDNA synthesis kit (Bio-rad, Hercules, CA, USA), following the manufacturer’s guidelines. KL-6 expression was assessed through quantitative real-time polymerase chain reaction (qRT-PCR) using SsoAdvanced™ SYBR^®^ Green Supermix (Bio-Rad).

Real-time PCR reactions were carried out in duplicate for each sample using 96-well plates and a reaction volume of 20 µL consisting of 1X SsoAdvanced ™ SYBR^®^ Green Supermix, 250 nM specific primers, and 100 ng of cDNA. The reactions were performed on the ABI-PRISM 7300 instrument (PE Applied Biosystems, Waltham, MA, USA). Relative quantification was conducted using the comparative 2^−ΔΔCt^ method [27], with β-actin serving as a normalizer [28]. The primer sequences employed for qRT-PCR amplification are detailed in Table 1.

### 2.7. Statistical Analysis

Results, denoted as mean ± standard deviation (SD), were subjected to assessment using Mann–Whitney’s U test to discern differences between groups. The interplay between KL-6 expression and key respiratory parameters was explored via Spearman’s correlation, scrutinizing both serum and broncho-alveolar lavage (BAL) samples. Survival analysis, contingent on KL-6 expression in both biological samples, was conducted through Kaplan–Meier curves. A *p* value of <0.05 was deemed statistically significant.

These analyses were meticulously executed employing GraphPad Prism software (version 9.0, GraphPad Software).

This comprehensive statistical approach ensures a robust examination of the relationships and prognostic implications associated with KL-6 expression in the context of pulmonary fibrosis.

## 3. Results

### 3.1. Demographic and Clinical Characteristics of the Population

The demographic and clinical characteristics of the 97 selected patients are listed in Table 2. A total of 39 patients with various forms of progressive pulmonary fibrosis, as well as 58 subjects with non-progressive fibrosis, with similar distributions of age, smoking, and gender ratio, were enrolled in this study (Figure 1) (Table 2). No correlations were found with respiratory data, and the Kaplan–Meier survival curve yielded statistically significant results.

### 3.2. KL-6 Gene Expression

KL-6 was expressed in all samples analyzed (Figure 2). Specifically, in BAL, the analysis of KL-6 expression showed statistically significant differences between the progressive fibrosis group, where it appears to be more highly expressed, and the nonprogressive fibrosis group, where, on the other hand, it is significantly reduced (*p* ≤ 0.0001) (Figure 2B). This difference was also confirmed at the serum level; in this case, too, there was a substantial difference between the progressive fibrosis group and the non-progressive fibrosis group (*p* = 0.0295) (Figure 2A). Therefore, in both BAL and serum, KL-6 values at diagnosis seem to be correlated with disease progression.

### 3.3. KL-6: BAL vs. SERUM

A significant difference was found by making a comparison between BAL and serum within the “progressive” group (Figure 3A). Notably, progressive pulmonary fibrosis was associated with markedly higher serum KL-6 expression than broncho-alveolar lavage (*p* = 0.0146).

KL-6, therefore, is more expressed both in the serum and in the BAL of patients affected by progressive pulmonary fibrosis than in patients diagnosed with non-progressive fibrosis; and, on a statistical level, among the expression levels of KL-6 found in patients with PPF, there is a significant difference between the BAL (local level) and the serum (systemic level).

The same holds true for patients who have non-progressive pulmonary fibrosis (Figure 3B). Again, KL-6 levels are higher in serum than in BAL (*p* = 0.0190).

## 4. Discussion

This study delves into the exploration of Krebs von den Lungen-6 (KL-6) as a potential biomarker to assess the severity and progression of interstitial lung diseases (ILDs), with a particular focus on progressive pulmonary fibrosis (PPF) compared to non-progressive (nPPF). Specifically, this study found that KL-6 expression was significantly higher in both the bronchoalveolar lavage (BAL) and serum of PPF patients compared to nPPF. This observation aligns with existing literature suggesting that elevated levels of KL-6 in serum correlate with a more severe prognosis and accelerated deterioration of ILDs, particularly idiopathic pulmonary fibrosis (IPF) [21]. Recent investigations have also confirmed the presence of higher levels of circulating KL-6 in various ILD subtypes compared to healthy control subjects, indicating a relationship between KL-6 and the presence of these lung diseases [29,30,31,32,33]. A key aspect of this study is the comparison between KL-6 levels in BAL and serum. The significant difference observed between the two indicates a notable variation in KL-6 expression at both local (BAL) and systemic (serum) levels. This disparity emphasizes the potential diagnostic and prognostic relevance of KL-6, providing valuable insights into both the local and systemic aspects of ILDs. This finding raises intriguing questions about the dynamics of KL-6 production, release, and clearance in different compartments of the respiratory system. The connection between serum KL-6 levels and the extent of pulmonary fibrosis has been widely demonstrated [34]. An increase in KL-6 levels can be interpreted as a signal of the presence and severity of lung damage, indicating increased alveolar-capillary membrane permeability. Essentially, KL-6 appears to be associated with critical indicators of lung health [20]. Previous studies have emphasized a link between serum KL-6 levels and ILD severity. A concrete example of this relationship was highlighted in a study that found a significant and inverse correlation between serum KL-6 levels and respiratory parameters such as the percentage of forced expiratory volume in one second (FEV1) and forced vital capacity. This suggests that higher KL-6 levels are associated with greater impairment of lung function [35]. Furthermore, another study has contributed to supporting the importance of circulating KL-6 levels in ILD diagnosis. This study represents the first meta-analysis to gather and synthesize the results of various studies examining the diagnostic and prognostic predictive values of KL-6. In practice, this indicates that KL-6 levels can be valuable indicators in assessing severity and diagnosing ILDs [36]. In summary, increased KL-6 levels appear to be associated with greater lung impairment, and their measurement may be useful for understanding and diagnosing these diseases. These findings further consolidate the critical role of KL-6 as a valuable indicator in the assessment of ILDs, providing not only an association with disease severity but also a diagnostic and predictive perspective. The integration of such results into our analysis strengthens the significance of KL-6 as a versatile biomarker with key implications for the clinical management of ILDs. The potential of KL-6 to serve as a guide in monitoring disease progression and treatment response underscores its clinical utility. This aligns with broader efforts in medical research to identify reliable biomarkers that can guide therapeutic strategies and improve patient outcomes. However, this study has some limitations. First, the number of patients included in this study may be relatively small. Second, this study did not assess the correlation between KL-6 levels and treatment response. Future studies with larger numbers of patients and longer follow-ups are needed to confirm the findings of this study and to assess the potential clinical utility of KL-6.

While this study provides valuable insights, challenges and avenues for future research remain. The heterogeneous nature of ILDs poses a challenge to standardizing diagnostic and prognostic approaches. Additionally, longitudinal studies are essential to validate the prognostic value of KL-6 over time. Exploring the underlying mechanisms of KL-6 in the context of fibrosis could reveal new therapeutic targets. The findings of this study suggest that KL-6 is a promising biomarker for the diagnosis, prognosis, and monitoring of ILDs. Further studies are needed to assess the potential clinical utility of KL-6, but this biomarker has the potential to improve the diagnosis and management of this disease.

## 5. Conclusions

In light of the results obtained, we can affirm that the ongoing quest for specific diagnostic biomarkers of progression in BAL is showing promising outcomes. The necessity for identifying progression markers arises from the diverse clinical trajectories observed in patients with interstitial lung diseases (ILDs), ranging from gradual and slow progression to rapid decline with early mortality or periods of relative clinical stability. Identifying patients at high risk of progression at the time of diagnosis is challenging, necessitating the exploration of predictive models based on clinical or biochemical parameters, such as KL-6. In our study, the exploration of progression biomarkers in serum and BAL reaffirmed the significance of serum KL-6 as a marker not only for the severity of the fibrosing process but also for disease progression. Simultaneously, it underscored that KL-6 in BAL could also serve as a promising biomarker of progression.

## Figures and Tables

**Figure 1 biomedicines-12-00269-f001:**
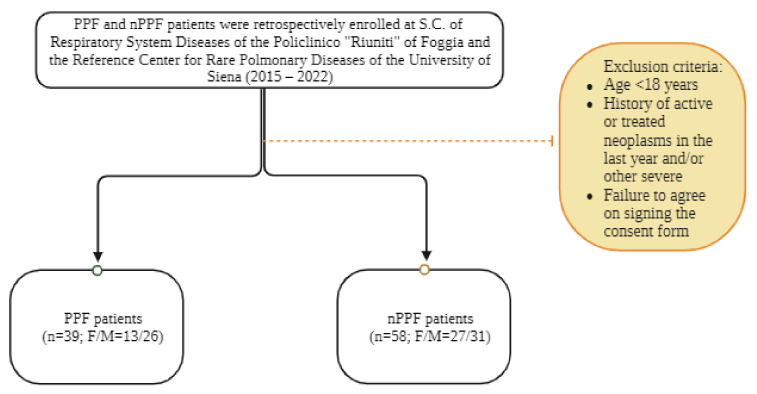
Flow chart of this study.

**Figure 2 biomedicines-12-00269-f002:**
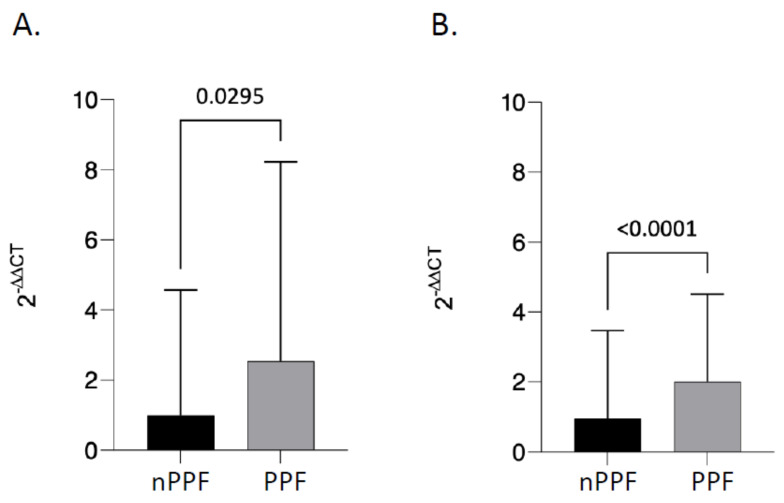
KL-6 gene expression. Quantitative real-time PCR analysis of KL-6 gene expression in progressive fibrosis (PPF) compared to patients with nonprogressive fibrosis (nPPF) both in serum (**A**) and BAL (**B**). β-actin was used as endogenous control. Comparisons between groups were performed by Mann–Whitney’s U test. Data are reported as mean ± SD. *p* ≤ 0.05.

**Figure 3 biomedicines-12-00269-f003:**
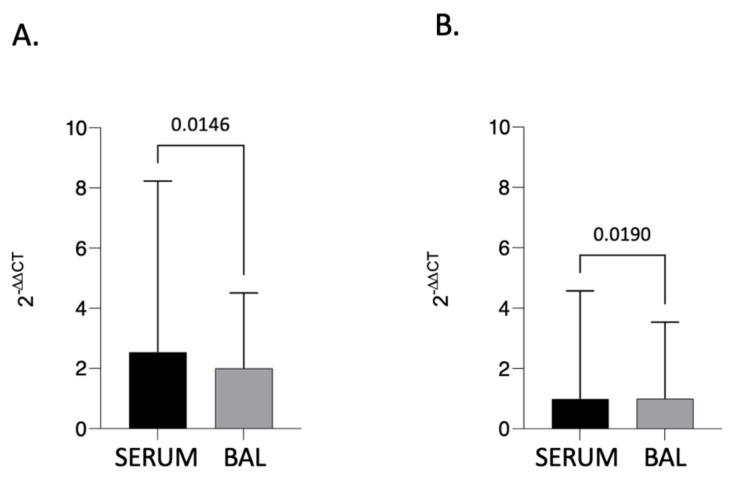
BAL vs. SERUM qRT-PCR analysis of KL-6 gene expression in progressive fibrosis (**A**) vs. nonprogressive fibrosis (**B**) either in serum or BAL. β-actin was used as endogenous control. Comparisons between groups were performed by Mann–Whitney’s U test. Data are reported as mean ± SD. *p* ≤ 0.05. Data in Figure 3 are a reproduction of data presented in Figure 2.

**Table 1 biomedicines-12-00269-t001:** Primers sequences.

KL-6	FORWARD	5′AGACGTCAGCGTGAGTGATG 3′
	REVERSE	5′GACAGCCAAGGCAATGAGAT 3′
β-actin	FORWARD	5′GACGACATGGAGAAAATCTG 3′
	REVERSE	5′ATGATCTGGGTCATCTTCTC 3′

**Table 2 biomedicines-12-00269-t002:** Demographic and clinical data of patients.

	ALL *n* = 97	PPF *n* = 39	nPPF *n* = 58	*p*-Value
Demographic data				
sex, % male	58.8	66.7	53.5	ns
age, years	64.5	65.0	64.3	ns
smoke, %	16.5	24.3	11.7	ns
Functional respiratory data				
FEV1, %	76.3	69.8	81.3	*p* = 0.0156 *
FVC, %	74.5	67.4	80.2	*p* = 0.0051 **
DLCO, %	58.5	48.9	66.4	*p* = 0.0005 **
6MWT, meters	352.2	337.0	364.4	ns

Summary of the main clinical data observed in patients. Data from the PPF and nPPF groups are reported as mean ± SD, and the comparison between the two groups was analyzed using Mann–Whitney’s U test. * *p* < 0.05, and ** *p* < 0.01. Abbreviations: FEV1: forced expiratory volume in the first second; FVC: forced vital capacity; DLCO: diffusing capacity for carbon monoxide; 6MWT: 6-min walking test.

## Data Availability

The data presented in this study are available on request from the corresponding author.

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
