# Peer review of "Role of BAL and Serum Krebs von den Lungen-6 (KL-6) in Patients with Pulmonary Fibrosis"

_biomedicines, 2024, doi:10.3390/biomedicines12020269_

Round 1

Reviewer 1 Report

Comments and Suggestions for Authors

The article covers a very interesting and current topic. The topic is interesting and the paper is quite well written. Nevertheless, in my opinion, some parts need to be improved, I have some comments:

1) Abstract. Results: In serum KL-6 is more highly expressed in PPF than in non-progressive fibrosis. 22 This difference is even more significant in BAL. Therefore, it is clear that KL-6 values are related to 23 disease progression. Significant differences were found by making a comparison between BAL and 24 serum. KL-6 was markedly higher in serum than BAL. Please, add the most important statistically significant values to support data.

2) Abstract. Conclusions: This study identified the role 25 of KL-6 as a promising biomarker of both fibrosing process severity and disease progression; more- 26 over, it highlighted how, even in BAL, KL-6 can represent a promising biomarker or progression. Abstract might be beneficial to include a sentence that briefly summarizes the key findings of the study. This can provide readers with a quick overview of the research. 

3) 1. Introduction 30 Idiopathic Pulmonary Fibrosis (IPF) stands as a formidable challenge within the 31 realm of respiratory diseases, representing a chronic and progressive affliction of the 32 lungs with an elusive etiology. I suggest that you include some information in order to complete the manuscript Below you can find some works that could give useful ideas in expanding this part:

a- Molecular and Genetic Biomarkers in Idiopathic Pulmonary Fibrosis: Where Are We Now? Biomedicines 202311, 2796. https://doi.org/10.3390/biomedicines11102796

b- Epithelial-Mesenchymal Transition: A Major Pathogenic Driver in Idiopathic Pulmonary Fibrosis? Medicina (Kaunas). 2020 Nov 13;56(11):608. doi: 10.3390/medicina56110608.

c- Study on Potential Differentially Expressed Genes in Idiopathic Pulmonary Fibrosis by Bioinformatics and Next-Generation Sequencing Data Analysis. Biomedicines 202311, 3109. https://doi.org/10.3390/biomedicines11123109

4) In so doing, we endeavored to cast a spotlight on any differentials manifesting at 92 local and systemic strata, underpinning KL-6's credentials as an auspicious and novel bi- 93 omarker for delineating the trajectory of the ailment. This scholarly pursuit, buttressed by 94 the aforementioned assumptions, seeks not only to enrich our comprehension of pulmo- 95 nary fibrosis dynamics but also to bestow upon KL-6 the mantle of a versatile and potent 96 biomarker with implications extending across both BAL and serum analyses. As we nav- 97 igate the intricate labyrinth of pulmonary fibrosis, the quest for precision and ... Please, underline the aim of the study and the novelty of the paper.

5) The subdivision based on disease progression was formulated 113 in accordance with the ATS/ERS/JRS/ALT 2022 guidelines [1] and the criteria recently pro- 114 posed by Hambly et al. to define disease progression [19]. Please, underline that you perfom a retrospective study using the last guidelines and the criteria recently proposed by Hambly et al. to define disease progression.

6) 3. Results 191 3.1. Demographic and Clinical Characteristics of the Population. Please add a flow-chart to clarify the selection of study population.

7) 3.4. Discussion 233 Idiopathic pulmonary fibrosis (IPF) stands as a relentless and enigmatic pulmonary ail- 234 ment, characterized by chronic and progressive fibrosis of the lung. The etiology remains 235 elusive, particularly prevalent in adulthood, and manifests through fibrotic changes in the 236 small interstitium, resulting in diminished lung volumes, compromised gas exchange, 237 and a relentless march towards respiratory failure culminating in mortality .. The discussion section needs to be improved.  It could be interesting to record the aim of the study. It is necessary to be more concise in the presentation of the facts, clarifying the results obtained and comparing them with previous or similar studies. However, it is interesting to answer the questions that arise from these results, backed up by published literature.

8) 4. Conclusions 270 In light of the results obtained, we can affirm that the search for specific diagnostic 271 biomarkers of progression in BAL is currently yielding good results. As previously ex- 272 plained, the need to identify progression markers stems from the considerable variety of 273 the clinical course of the patient affected by IPF who may undergo a gradual and slow 274 progression or a very rapid decline with early mortality as well as periods of relative clin- 275 ical stability ... Please, underline the novelty of the study and the limitations of the study.

Comments on the Quality of English Language

Minor changes of English language are required

Reviewer 2 Report

Comments and Suggestions for Authors

This is study assesses KL-6 levels in serum and BAL. The authors do a nice job placing their findings in the context of lung fibrotic disease; however, certain aspects of data evaluation, presentation, and interpretation require further clarification. Specific comments are detailed below:

 Specific Comments:

Major

1)      In table 2, it is listed that the * symbol relates to p ≤ 0.05. However, this symbol is not used in the table. Furthermore, it is not specified what the p values listed in the table relate to. Is it the p value for PPF vs. nPPF? This can be clarified in the legend.

2)      The data in figures 1 and 2 appear redundant. Either all comparisons should be indicated in 1 figure or it should be stated explicitly in the legend that data in figure 2 are a replot of data in figure 1.  It may also be beneficial to include the preparation in the y axis [for example 2-ΔΔCT (BAL) and 2-ΔΔCT (Serum)] for clarity. As you list specific p values, the * symbol in the legend is unnecessary.

3)      Although the authors show significant differences for the serum and BAL levels of KL-6 (Figure 2), the effect size is modest for PPF (2A), and almost nonexistent in nPFF (2B). It is not clear what the authors are trying to say about this other than BAL serves as an additional way to measure KL-6.

4)      No data for KL-6 levels are provided for non-IPF patients. It is understood that getting non-diseased patients to participate in these invasive procedures may not be possible, however, some discussion of what levels would be like in these non-diseased patients would be beneficial to the manuscript.

5)      The authors suggest (and the data support) that KL-6 levels are a potential indicator of PPF or nPPF. However, it is not clear how KL-6 levels may be changed over time in PPF, which may limit the interpretation of these results, i.e. are the higher levels just an artifact of the most severe cases of PPF? It would be beneficial to include plots of FEV, FVC and DLCO vs. KL-6 levels and determine whether or not there are correlations between these variables. If not, it would greatly strengthen the conclusion that KL-6 serves as a predictive marker and not an effect of additional damage.

Minor

1)      In line 175 (methods, RT-PCR) it appears that the “ΔΔ” symbols are missing.

2)      Should table 1 read “β-actin” instead of “-actina”?

Round 2

Reviewer 1 Report

Comments and Suggestions for Authors

The manuscript has been substantially modified in all its parts. Thank you for responding to all concerns raised carefully and thoroughly. I have no further comments.

Reviewer 2 Report

Comments and Suggestions for Authors

The authors have sufficiently responded to reviewer concerns.

Comments on the Quality of English Language

A few editorial errors remain (that can easily be addressed during publication process), but the overall quality of English has been improved since the 1st submission.